# Alterations in p53, Microsatellite Stability and Lack of MUC5AC Expression as Molecular Features of Colorectal Carcinoma Associated with Inflammatory Bowel Disease

**DOI:** 10.3390/ijms24108655

**Published:** 2023-05-12

**Authors:** Míriam Gené, Míriam Cuatrecasas, Irene Amat, Jesús Alberto Veiga, María Jesús Fernández Aceñero, Victòria Fusté Chimisana, Jordi Tarragona, Ismael Jurado, Rebeca Fernández-Victoria, Carolina Martínez Ciarpaglini, Cristina Alenda González, Carlos Zac, Pilar Ortega de la Obra, María Teresa Fernández-Figueras, Manel Esteller, Eva Musulen

**Affiliations:** 1Pathology Department, Hospital Universitari Joan XXIII, 43005 Tarragona, Spain; miriamgehi@hotmail.com; 2Surgery Department, Programme of Surgery and Morphological Sciences, Universitat Autònoma de Barcelona (UAB), 08193 Barcelona, Spain; 3Pathology Department, Hospital Clínic de Barcelona, Universitat de Barcelona (UB), 08007 Barcelona, Spain; mcuatrec@clinic.cat; 4School of Medicine, Campus Clínic, Universitat de Barcelona (UB), 08036 Barcelona, Spain; 5Pathology Department, Complejo Hospitalario de Navarra, 31008 Navarra, Spain; irene.amat.villegas@navarra.es; 6Pathology Department, Complejo Hospitalario Universitario de Ferrol, 15405 Ferrol, Spain; jesus.alberto.veiga.barreiro@sergas.es; 7Pathology Department, Hospital Clínico San Carlos, 28040 Madrid, Spain; mgg10167@gmail.com; 8Pathology Department, Hospital de la Santa Creu i Sant Pau, 08025 Barcelona, Spain; vfuste@santpau.cat; 9Pathology Department, Hospital Universitari Arnau de Vilanova, 25198 Lleida, Spain; jtarragona@gss.cat; 10Pathology Department, Consorci Sanitari de Terrassa, 08227 Terrassa, Spain; ijurado@cst.cat; 11Pathology Department, Hospital Álvaro Cunqueiro, 36312 Vigo, Spain; rebeca.fernandez.victoria@sergas.es; 12Pathology Department, Hospital Clínico Universitario de Valencia, Valencia INCLIVA-Instituto de Investigación Sanitaria, Universidad de Valencia, 46010 Valencia, Spain; carolaciarpaglini@gmail.com; 13Pathology Department, Hospital General Universitario Dr. Balmis, Instituto de Investigación Sanitaria y Biomédica de Alicante (ISABIAL), 03010 Alicante, Spain; alenda.cris@gmail.com; 14Pathology Department, Hospital Universitari i Politècnic La Fe, 46026 Valencia, Spain; carloszacromero@gmail.com; 15Pathology Department, Hospital General de Segovia, 40002 Segovia, Spain; mportegao@saludcastillayleon.es; 16Pathology Department, Hospital Universitari General de Catalunya-Grupo QuironSalud, Sant Cugat del Vallès, 08195 Barcelona, Spain; maite.ffigueras@quironsalud.es; 17School of Medicine, Campus Sant Cugat del Vallès, Universitat Internacional de Catalunya (UIC), Sant Cugat del Vallès, 08017 Barcelona, Spain; 18Institut de Recerca contra la Leucèmia Josep Carreras (IJC), Badalona, 08916 Barcelona, Spain; mesteller@carrerasresearch.org; 19Institució Catalana de Recerca i Estudis Avançats (ICREA), 08010 Barcelona, Spain; 20Faculty of Medicine and Health Sciences, Department of Physiological Sciences, Universitat de Barcelona (UB), 08007 Barcelona, Spain; 21Centro de Investigación Biomédica en Red Cáncer (CIBERONC), 28029 Madrid, Spain

**Keywords:** intestinal bowel disease, colorectal cancer, MUC5AC expression, gastric metaplasia, DNA-damage response

## Abstract

Colitis-associated colorectal carcinoma (CAC) occurs in inflammatory bowel disease (IBD) because of the “chronic inflammation-dysplasia-cancer” carcinogenesis pathway characterized by p53 alterations in the early stages. Recently, gastric metaplasia (GM) has been described as the initial event of the serrated colorectal cancer (CRC) process, resulting from chronic stress on the colon mucosa. The aim of the study is to characterize CAC analyzing p53 alterations and microsatellite instability (MSI) to explore their relationship with GM using a series of CRC and the adjacent intestinal mucosa. Immunohistochemistry was performed to assess p53 alterations, MSI and MUC5AC expression as a surrogate for GM. The p53 mut-pattern was found in more than half of the CAC, most frequently stable (MSS) and MUC5AC negative. Only six tumors were unstable (MSI-H), being with p53 wt-pattern (*p* = 0.010) and MUC5AC positive (*p* = 0.005). MUC5AC staining was more frequently observed in intestinal mucosa, inflamed or with chronic changes, than in CAC, especially in those with p53 wt-pattern and MSS. Based on our results, we conclude that, as in the serrated pathway of CRC, in IBD GM occurs in inflamed mucosa, persists in those with chronic changes and disappears with the acquisition of p53 mutations.

## 1. Introduction

Crohn’s disease (CD) and ulcerative colitis (UC) are pathologies with a two- to three-fold increased risk of colorectal cancer (CRC) [1,2]. Likewise, colitis-associated colorectal cancer (CAC) is the leading cause of death among long-standing inflammatory bowel disease (IBD) patients [3]. The continuous inflammatory stress of the colonic mucosa, with constant epithelial lesion repair and alterations of the microbiota, damage the colonic epithelia leading to dysplasia. As a result, CAC appears following the same pattern of distribution as inflammation in the different IBD subtypes. In CD, neoplasms are most often located in the small bowel or on the right side of the colon, whereas in UC tumors are usually located in the rectum and left colon. CACs are more frequently poorly differentiated, mucinous or signet ring cell carcinomas [4]. Recently, similar characteristics between CAC and CRC with microsatellite instability high (MSI-H), whether sporadic MSI-H or Lynch syndrome (LS)-related, have been described [5,6,7].

The sequence “chronic inflammation-dysplasia-cancer” should determine the driving genes that guide this carcinogenesis pathway of CRC, which is different from the canonical and serrated pathways [8]. In this “inflammatory” context the Wnt-signaling pathway is not involved, and *TP53* alterations appear early and with different hotspot mutations as opposed to the canonical pathway where *TP53* plays a role at later stages [9,10,11,12]. In addition, copy number alterations are frequently found involving genes such as *c-MYC*, *IRS2* and *HER2* as well as mutations in *BRAF* and *IDH1* and the activation of transforming growth factor β related to the epithelial-mesenchymal transition [10,13,14]. In LS, the mutation of genes of the mismatch repair (MMR) system (*MLH1*, *MSH2*, *MSH6* and *PMS2*) are the hallmark of this hereditary cancer predisposition syndrome [15]. In the serrated pathway, the mutation of genes in the MAP kinases cascade, especially *BRAF*, and the methylation of CpG islands are the main effectors [16].

In CD the presence of metaplastic changes such as pyloric gland metaplasia or gastric foveolar metaplasia have been described in the small bowel mucosa or in pouchitis in long-standing disease [17,18,19,20,21,22]. Recently, gastric metaplasia (GM) has been identified as a substrate for the initiation of the serrated pathway of CRC, due to the continuous lesion of the colonic epithelium of the right colon [23]. Hence, considering that both the serrated and IBD-associated pathways share the sequence “inflammation-dysplasia-carcinoma”, we hypothesize that they could have a common origin in GM, as a mechanism of adaptation to colonic mucosal stress.

The aims of the study are to characterize CAC analyzing p53 alterations and microsatellite instability (MSI) and to explore their relation to GM.

## 2. Results

### 2.1. Patients and Specimens

A total of 57 CACs were studied from 38 (72%) men and 15 (28%) women, all Caucasian, aged between 34 and 90 years (mean 62.6), 34 (64%) with UC and 19 (36%) with CD.

According to location, CAC in patients with CD were more frequently located in the ileum (7/19, 37%) and the right colon (6/19, 31.5%), whereas CAC in patients with UC were more frequent in the left colon (15/38, 39.5%). This distribution was statistically significant (*p* = 0.003). All eight (21%) synchronous tumors were found in UC which was statistically significant (*p* = 0.042). In most cases (29/53, 55%) the colectomy mucosa showed active inflammation. Regarding histology, the majority of CRC (47/57, 82.5%) were adenocarcinomas in both CD and UC. Intestinal mucosa adjacent to the tumor was identified in 35 (61.4%) cases mostly with chronic changes, especially in UC. According to staging, stage III was more frequent in CD (10/19, 52.6%) while in UC the most prevalent was stage II (17/38, 44.7%) (Table 1).

### 2.2. Immunohistochemistry

#### 2.2.1. p53, MUC5AC and MSI in Adenocarcinoma

The p53-mut pattern (defined in Section 4) was found in more than half (29/57, 51%) of the adenocarcinomas (22 with overexpression and 7 with null pattern). The differences in the distribution of both expression patterns between CD and UC were not statistically significant (Table 2).

When MSI was analyzed, we identified only six (11%) tumors that were unstable (MSI-H), all with a lack of expression for MLH1/PMS2, with the same distribution in both IBD subtypes (Table 2).

MUC5AC staining was identified in 44% of CACs, 9 adenocarcinomas with positivity in more than 50% of the tumor (extensive expression) and 16 with focal expression. The differences in the distribution of both expression patterns between CD and UC were not statistically significant (Table 2).

#### 2.2.2. Relation of the Three IHC Markers in Adenocarcinoma

Analyzing p53 expression with MSI, we observed that the six unstable tumors were p53 wt-pattern (100%), and most of the stable tumors presented a p53 mut-pattern (29/51, 57%) establishing a statistically significant relation (*p* = 0.010) (Table 3).

When we analyzed p53 according to MUC5AC expression, we found that 60% of CACs with MUC5AC expression had a p53 wt-pattern. In contrast, 59% of CACs with p53 mut-pattern were MUC5AC negative. Nevertheless, this distribution was not statistically significant (Table 4 and Figure 1).

Analyzing the relation between MUC5AC expression and MSI, a statistically signification was found, with all MSI-H CACs being positive for MUC5AC and all MSS CACs being MUC5AC negative (Table 5).

The clinicopathological characteristics and IHC expression of MSI-H CAC are shown in Figure 2 and Table 6. Of notice, these adenocarcinomas were in the ileum or right colon and showed histological features characteristic of sporadic MSI-H or LS-associated CRC—i.e., medullary, mucinous or signet ring cell carcinomas.

Only one patient with MSH-H CAC was affected by LS, and another had two synchronous sporadic tumors harboring the V600E *BRAF* mutation. The remaining unstable CRCs were *BRAF* wild type, and the hypermethylation of *MLH1* was not performed.

#### 2.2.3. p53 and MUC5AC Expression in the Mucosa Adjacent to Adenocarcinomas

To determine whether p53 and MUC5AC staining were present in the non-tumoral intestinal mucosa, we explored the mucosa adjacent to the adenocarcinomas, which was available in 35 cases, 1 from the small intestine and 34 from the colon.

p53 immunostaining showed focal positivity in the nuclei of the glands surrounding the adenocarcinomas in all cases, corresponding to wt-pattern.

MUC5AC immune-expression was found in 77% (27/35) of the mucosa adjacent to CAC, being in the ileum (100%, 1/1), in 70% (7/10) of the mucosa located in the right colon, in 80% in the left colon (12/15) and in 100% (4/4) of the rectal mucosa. Regarding the type of lesion, MUC5AC was positive in all cases with active inflammation, in 79% (15/19) of the mucosa with chronic changes and in 67% (8/12) of the mucosa without morphological lesions, although these differences were not statistically significant (Figure 3).

The relation between MUC5AC expression in adenocarcinoma and adjacent mucosa was not statistically significant. MUC5AC-positive non-tumoral mucosa was adjacent to negative adenocarcinoma in more than 60% (16/27), and MUC5AC-negative mucosa was adjacent to positive or negative adenocarcinoma in the same proportion (50%, 4/8) (Table 7 and Figure 4).

## 3. Discussion

Colitis-associated colorectal cancer (CAC) is the result of the alteration of genes involved in the sequence of “chronic inflammation-dysplasia-cancer” which draw a different landscape from the other colorectal cancer pathways.

It is well known that p53 alterations appear in early stages even in the colon mucosa without dysplastic changes, contributing to establish a field cancerization [9,14]. In line with published data, in our series a pattern of p53 mutation was present in more than half of the CAC, independently of the IBD subtype [9,10,11,12,13,14]. However, not any of the p53 mut-patterns were found in any adjacent intestinal mucosa.

MSI-H in IBD, especially in UC, has been reported between 3 and 18% [5,6,7,24,25,26,27,28,29,30]. In our study, MSI-H was found in 10.5%, within the described range and with the same incidence in CD and UC. Furthermore, we found a statistical signification between p53 and MSI, such as that all p53 mut-pattern were stable tumors and all MSI-H CAC showed a p53 wt-pattern. Therefore, our results support a role for MSI in a subset of CAC. In addition, our unstable tumors had other clinical and pathological features of MSI-H CRC, such as the right location in the colon and histology of mucinous, medullary or signet ring cell carcinomas, in line with previous observations of other authors [5,6,7]. In our study, LS was excluded in one patient older than 70 years with UC and two synchronous tumors, since the molecular characteristics of both CRC and family history did not support LS diagnosis. Multifocal neoplasms are present in 38% of UC patients, and these synchronous neoplasms are usually heretogeneous [14]. In contrast, our patient had two MSI-H CACs which shared the molecular alterations and histological features probably due to both neoplasms appearing in the same side of the colon.

The other four patients with MSI-H CRC were in their 50s. One was a LS with UC of 4-year duration. The CRC was probably the result of the summatory risk of the two conditions. The presence of IBD and LS in the same patient is a rare situation given the scarce literature reporting these cases [31,32,33,34,35,36,37] and the low rate of germline mutations found in IBD patients [12]. Nevertheless, it is important to exclude LS in those MSI-H CACs to define the outcome in patients with early onset CRC, which is worse in the IBD-related group, intermediate in the sporadic group and best in the hereditary group [35]. In the remaining three patients with MSI-H CRC, short course IBD and tumors with wt-*BRAF*, LS cannot be ruled out.

MUC5AC staining, characteristic of gastric foveolar epithelium [38], was identified in less than half (44%) of the CACs but was retained in 77% of the 35 available adjacent intestinal mucosa. The MUC5AC-positive CACs were more frequently those with a p53 wt-pattern and MSS, although MUC5AC expression was present in all cases with MSI-H. Furthermore, considering that most of the tumors with p53 mut-pattern lack MUC5AC expression, it can be deduced that the appearance of gastric epithelial change occurs in the inflamed mucosa, persists in those with chronic-regenerative changes and is maintained in CAC without p53 mutation, to finally disappear in parallel to the acquisition of p53 mutations. Our results agree with the recently published data by Chen et al. who identified GM as a substrate for the initiation of the serrated pathway of CRC, due to continuous injury of the colonic epithelium of the right colon [23]. Therefore, we could argue that GM is the change shared by CAC and serrated CRC in which the sequence “chronic inflammation-dysplasia-cancer” is initiated. Considering all this, we cannot rule out that the group of MSI-H CRC that appear in IBD, with characteristics like MSI-H sporadic tumors with *MLH1* hypermethylation, are nothing more than serrated pathway CRCs that appear in a context of continuous colon injury.

Further studies are needed to explore the expression of MUC5AC in dysplastic epithelia IBD-associated, an intermediate step between colon mucosa with chronic changes and the onset of neoplastic events of CAC.

## 4. Materials and Methods

We retrospectively reviewed a series of 53 IBD colectomies with CAC from 14 hospitals. Specimens were classified as CD or UC based on histology, radiological studies and clinical information collected from patient histories. Intestinal mucosa adjacent to the tumor was evaluated for the presence of inflammatory activity or chronic changes.

### 4.1. Immunohistochemistry

Formalin-fixed, paraffin-embedded tissue sections were analyzed using standard IHC techniques. The primary antibodies used were anti-p53 (clone DO-7, Ventana Medical Systems, Inc., 1910 E. Innovation Park Drive, Tucson, AZ, USA), anti-MUC5AC (clone MRQ-19, Ventana Medical Systems), anti-MLH1 (clone ES05, Leica Biosystems, Newcastle Upon Tyne, UK), anti-MSH2 (clone 79H11, Leica Biosystems), anti-MSH6 (clone EP49, Leica Biosystems) and anti-PMS2 (clone EP51, Leica Biosystems, Newcastle Upon Tyne, UK). Immunostaining was performed automatically using a Ventana BenchMark ULTRA machine (Roche, Basel, Switzerland) for p53 and MUC5AC and a Bond-III machine (Leica Biosystems, Newcastle Upon Tyne, UK) for the MMR antibodies. Immunostaining was independently evaluated by two pathologists (M.G. and E.M.).

For the evaluation of p53 immunostaining, we used the 3-level scores described by Köbel et al. that correlated with *TP53* alterations: the overexpression pattern, when strong and intense p53 staining was observed in all nuclei, and the complete absence (null pattern), when a total absence of nuclear staining was observed, were considered as surrogates for *TP53* mutation (p53-mut pattern) [39]. The wild-type pattern, corresponding to focal nuclear positivity of varying intensity, was considered normal (p53-wt pattern), even knowing that it could be a false negative for a mutated *TP53* [39] (Figure 5). An external positive control was included on each slide.

Positive staining for the MMR antibodies were in the nucleus of the neoplastic cells. We considered the tumor stable if nuclear staining was retained in tumor cells for all four MMR antibodies and unstable if one or more of the four markers lacked nuclear expression in tumor cells. Nuclear immunoreaction in lymphocytes, normal colonic mucosa or stromal cells served as the internal anti-MMR positive control. Loss of MMR staining was considered when the nuclei of all neoplastic cells were negative.

MUC5AC in adenocarcinomas was considered positive extensively if more than 50% of the tumor glands showed positive staining in the cytoplasm, focally positive if less than 50% of the glands had expression in the cytoplasm and negative if 100% of the tumor glands lacked expression. MUC5AC in the mucosa adjacent to the CRC was consider positive if more than 50% of the gland showed cytoplasmic staining. An external positive control was included on each slide.

Microscopy images were acquired with a digital slide scanner (Ultra Fast Scanner 1.8, Philips, Amsterdam, The Netherlands).

### 4.2. Statistical Analysis

Analysis was carried out using SSPS software version 26.0 (SPSS, Chicago, IL, USA). The χ2 test was used to analyze the association between qualitative variables, whereas the Fisher’s exact test and the Student’s *t*-test or the Mann–Whitney test were used for quantitative variables. A *p* < 0.05 was considered statistically significant.

## 5. Conclusions

Our results reflect that CAC shows a molecular profile characterized by p53 alterations and an absence of both MSI and MUC5AC staining, which differentiate it from CRC of the canonical and serrated carcinogenesis pathways. In relation to the expression of MUC5AC in CAC and surrounding intestinal mucosa, we could hypothesize that the appearance of GM occurs in inflamed mucosa, persists in those with chronic-regenerative changes and disappears with the acquisition of p53 mutations (Figure 6).

## Figures and Tables

**Figure 1 ijms-24-08655-f001:**
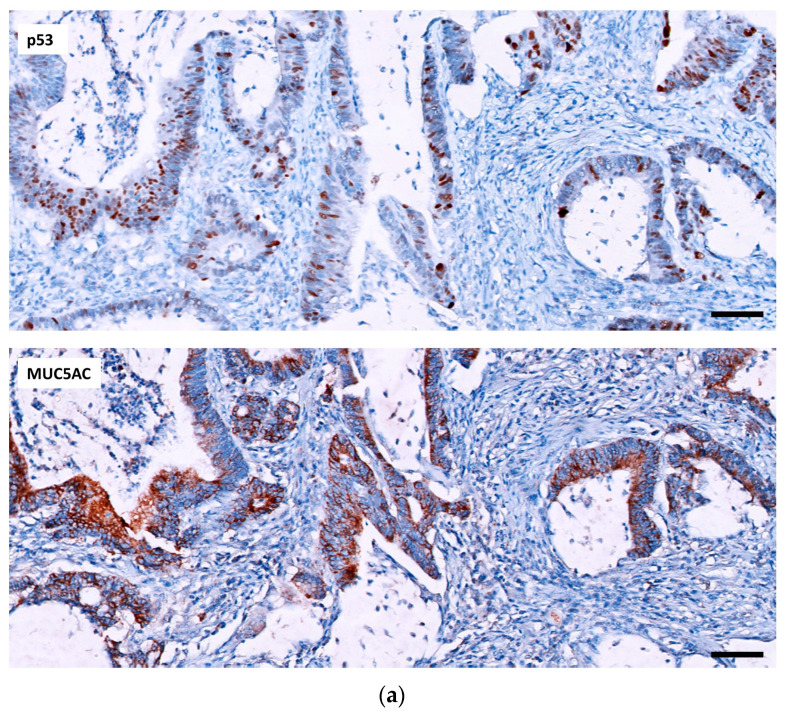
Two CACs with different expression of p53 and MUC5AC. (**a**) p53 shows a wt-pattern with focal nuclear staining in tumor cells. In the same tumor, MUC5AC expression is diffuse (20×, scale bar 50 µm). (**b**) In this other tumor, nuclear overexpression is observed in the form of mut-pattern staining of p53 while MUC5AC is negative. Note the MUC5AC positivity in non-dysplastic glands on the left side (10×, scale bar 100 µm).

**Figure 2 ijms-24-08655-f002:**
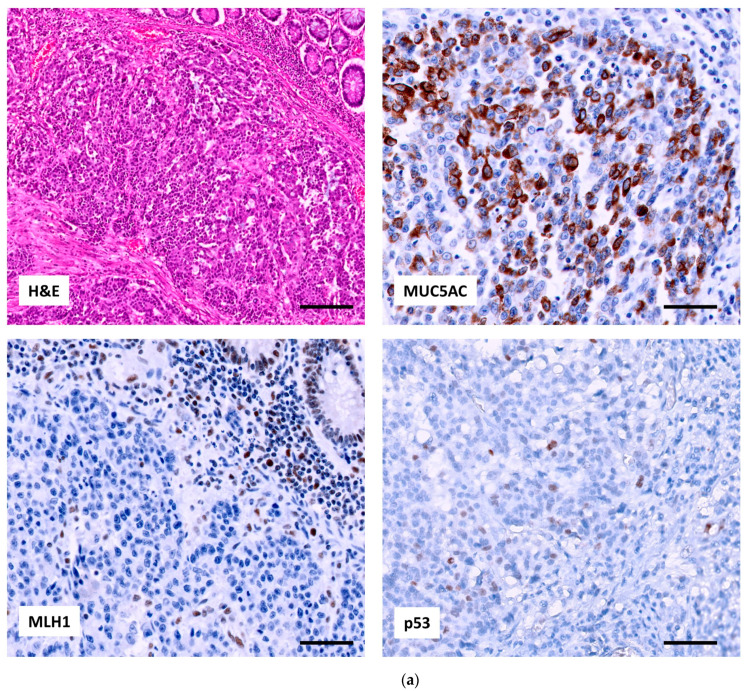
Two examples of MSI-H CAC. (**a**) medullary and (**b**) mucinous. In both adenocarcinomas p53 showed scattered nuclear positivity corresponding to a wt-pattern, MUC5AC expression was focal positive, and MLH1 showed an absence of nuclear staining in tumor cells. Note in both MLH1 staining the positive internal control in lymphocytes (20×, scale bar 50 µm).

**Figure 3 ijms-24-08655-f003:**
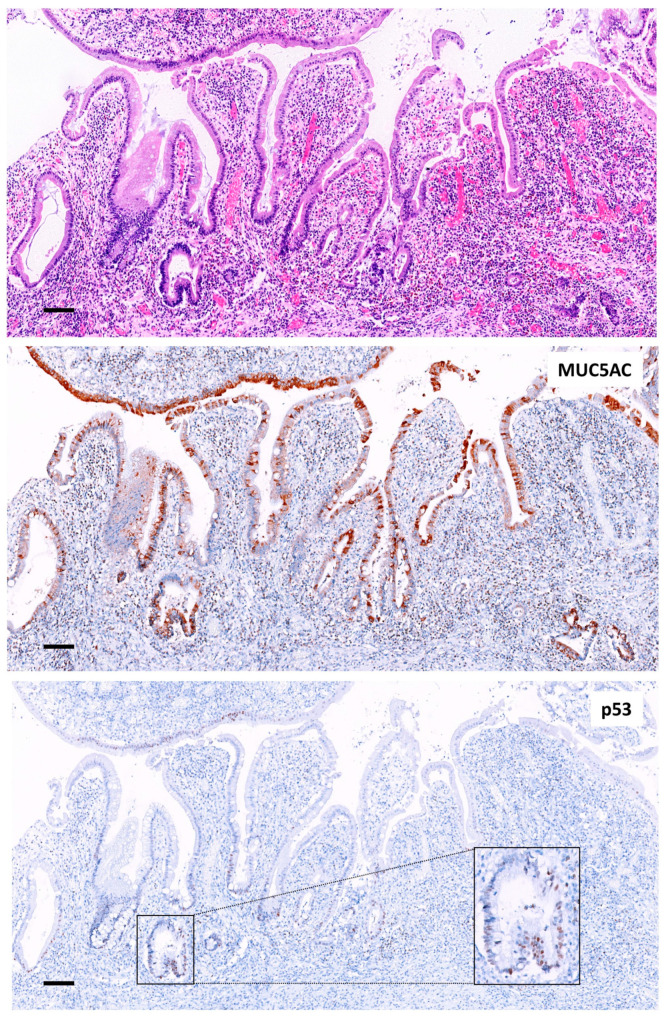
Colon mucosa adjacent to a CAC. Severe distortion of colonic epithelia with chronic inflammation, glandular atrophy and granulation tissue (H&E, 10×, scale bar 100 µm). Strong expression of MUC5AC was seen in the epithelium re-epithelizing the granulation tissue and in part of the irregular glands (10×, scale bar 100 µm). p53-wt pattern showed few positive nuclei in the surface epithelium and in the glands (10×, scale bar 100 µm; in magnification, 40×).

**Figure 4 ijms-24-08655-f004:**
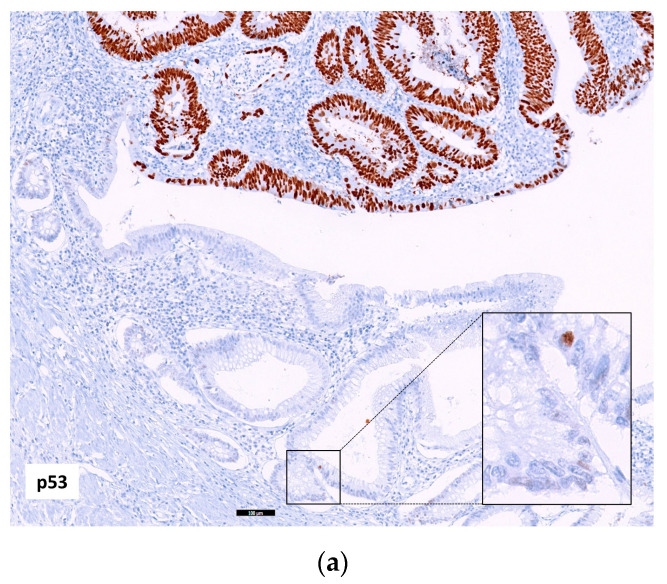
Colonic mucosa with chronic changes adjacent to a CAC. (**a**) p53 staining showed a wt-pattern with few positive nuclei (in magnification) in the colon glands adjacent to an adenocarcinoma with p53 nuclei overexpression (p53 mut-pattern) (10×, scale bar 100 µm). (**b**) MUC5AC strongly positive in the colon mucosa adjacent to a negative neoplasm (5×, scale bar 200 µm).

**Figure 5 ijms-24-08655-f005:**
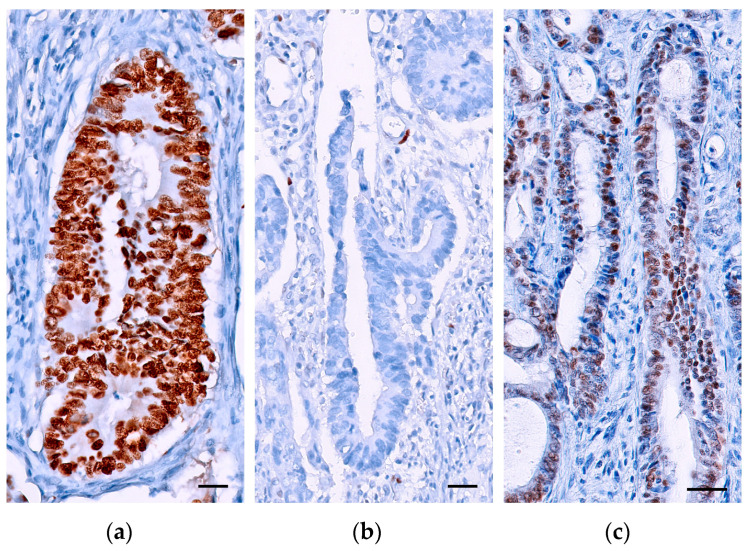
Three patterns of p53 staining. (**a**) Overexpression pattern, all tumor nuclei are strongly positive (scale bar 10 µm). (**b**) Null pattern, no staining is observed in tumor cell nuclei in the presence of a positive internal control in stromal cells (scale bar 10 µm). (**c**) Wild-type pattern, focal nuclear positivity of varying intensity in tumor cell nuclei (scale bar 50 µm).

**Figure 6 ijms-24-08655-f006:**
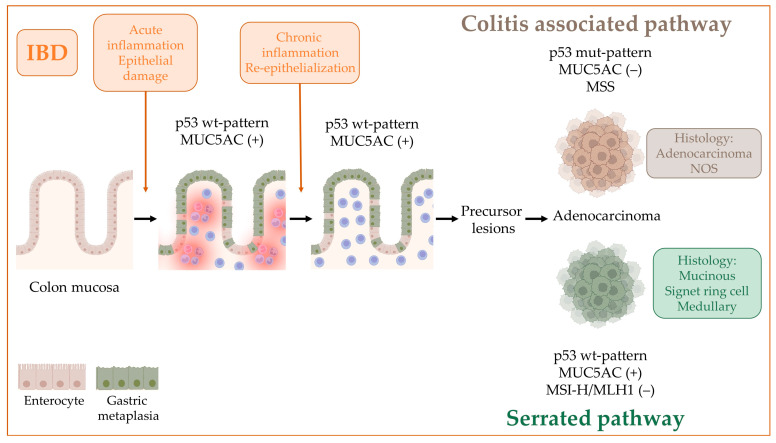
Pathways of CRC carcinogenesis in IBD. GM appears in the inflamed colic mucosa as an adaptive mechanism to continued stress, persists in the mucosa with chronic changes and disappears with the acquisition of p53 mutations. CACs are characterized by p53 alterations, a lack of MUC5AC expression and MSS, whereas those CRCs with MSI-H, MUC5AC expression and p53 wt-pattern could represent sporadic CRC of the serrated pathway that appear in a context of “chronic inflammation-dysplasia-cancer” from a gastric metaplastic epithelium. Figure created with BioRender.com.

**Table 1 ijms-24-08655-t001:** Clinicopathological features of IBD patients and adenocarcinomas.

	Crohn’s Disease (%)	Ulcerative Colitis (%)	Total (%)	*p* Value
Patients	19 (36)	34 (64)	53 (100)	
Age Mean years [range]	58.4 [35–88]	64.8 [34–90]	62.6 [34–90]	
Gender				1.000
Male	14 (74)	24 (71)	38 (72)	
Female	5 (26)	10 (29)	15 (28)	
IBD duration				0.244
Mean years [range]			13.2 [1–43]	
Concominant *	5 (26)	2 (6)	7 (13.2)	
≤8 years	4 (21)	13 (38)	17 (32.1)	
9–15 years	3 (16)	7 (20.6)	10 (18.9)	
16–20 years	1 (5)	5 (14.7)	6 (11.3)	
>21 years	5 (26)	6 (17.6)	11 (20.8)	
Unknown	1 (5)	1 (3)	2 (3.8)	
IBD activity				0.780
Active	11 (58)	18 (53)	29 (55)	
Quiescent	8 (42)	16 (47)	24 (45)	
Synchronous CRC (*n* = 57)				0.042
Non synchronous CRC	19 (100)	30 (79)	49 (86)	
2 synchronous CRC	0	8 (21)	8 (14)	
Adenocarcinoma location (*n* = 57)				0.003
Ileum	7 (37)	0	7 (12.3)	
Right colon	6 (31.5)	11 (28.9)	17 (29.8)	
Left colon	3 (16)	15 (39.5)	18 (31.6)	
Rectum	1 (5)	6 (15.8)	7 (12.3)	
Colon	2 (10.5)	5 (13.2)	7 (12.3)	
Ileal reservory	0	1 (2.6)	1 (1.8)	
Histology (*n* = 57)				0.504
Adenocarcinoma NOS	15 (78.9)	32 (84.2)	47 (82.5)	
Mucinous	3 (15.8)	4 (10.5)	7 (12.3)	
Mucinous and signet ring cell	1 (5.3)	0	1 (1.8)	
Medullary	0	1 (2.6)	1 (1.8)	
Undifferentiated	0	1 (2.6)	1 (1.8)	
Histologic grade (*n* = 57)				1.000
Low-grade	15 (79)	31 (82)	46 (81)	
High-grade	4 (21)	7 (18)	11 (19)	
Adjacent mucosa to adenocarcinoma (*n* = 57)				0.077
Absent	10 (53)	12 (32)	22 (39)	
Present	9 (47)	26 (68)	35 (61)	
Active inflammation	0	4 (15)	4 (11.4)	
Chronic changes	4 (21)	15 (58)	19 (54.3)	
Normal	5 (26)	7 (27)	12 (34.3)	
Stage (8th ed) (*n* = 57)				0.305
I	3 (15.8)	9 (23.7)	12 (21)	
II	6 (31.6)	17 (44.7)	23 (40)	
III	10 (52.6)	12 (31.6)	22 (39)	

Abbreviations: IBD, intestinal bowel disease; NOS, not otherwise specified. Concomitant*, in this case the diagnosis of IBD was made at colectomy.

**Table 2 ijms-24-08655-t002:** Expression of p53, MSI and MUC5AC in CAC according to IBD subtypes.

		Crohn’s Disease (%)	Ulcerative Colitis (%)	Total (%)	*p* Value
p53 pattern					0.914
mut-pattern	Overexpression	8 (42)	14 (37)	22 (39)	
	Null pattern	2 (11)	5 (13)	7 (12)	
wt-pattern	Focal nuclear staining	9 (47)	19 (50)	28 (49)	
Microsatellite instability					0.389
	MSI-H	3 (16)	3 (8)	6 (11)	
	MSS	16 (84)	35 (92)	51 (89)	
MUC5AC					0.511
Negative		10 (52.6)	22 (57.9)	32 (56.1)	
Positive	Extensive	2 (10.5)	7 (18.4)	9 (15.8)	
	Focal	7 (36.8)	9 (23.7)	16 (28.1)	

Abbreviation: mut, mutation; wt, wild type; MSI-H, microsatellite instability high grade; MSS, microsatellite stable.

**Table 3 ijms-24-08655-t003:** Relation between p53 immunohistochemical patterns and MSI.

	Microsatellite Instability			
	MSI-H (%)	MSS (%)	Total (%)	*p* Value
p53 pattern				0.010
mut-pattern	0	29 (57)	29 (51)	
wt-pattern	6 (100)	22 (43)	28 (49)	

Abbreviation: MSI-H, microsatellite instability high-grade; MSS, microsatellite stable; mut, mutation; wt, wild type.

**Table 4 ijms-24-08655-t004:** Relation of p53 and MUC5AC expression in CAC.

	MUC5AC			
	Negative (%)	Positive (%)	Total (%)	*p* Value
p53 pattern				0.186
mut-pattern	19 (59)	10 (40)	29 (51)	
wt- pattern	13 (41)	15 (60)	28 (49)	

Abbreviation: mut, mutation; wt, wild type.

**Table 5 ijms-24-08655-t005:** Relation between MUC5AC expression and MSI.

	MUC5AC			
	Negative (%)	Positive (%)	Total (%)	*p* Value
Microsatellite instability				0.005
MSI-H	0	6 (24)	6 (11)	
MSS	32 (100)	19 (76)	51 (89)	

Abbreviation: MSI-H, microsatellite instability high-grade; MSS, microsatellite stable.

**Table 6 ijms-24-08655-t006:** Clinicopathological and IHC features of unstable CRC.

Patient	Age (Years)	Gender	IBD	IBD Duration (Years)	Location	Histology	p53 Pattern	MUC5AC	MLH1	*BRAF* Status	*MLH1* Methylation	Predisposition to Cancer
8	73	Male	UC	12	Right colon	Medullary	wt	Focal positive	Negative	V600E	NP	Sporadic vs. IBD-associated
8					Right colon (transvers)	NOS with mucinous component	wt	Focal positive	Negative	V600E	NP	Sporadic vs. IBD-associated
14	52	Male	UC	4	Right colon	NOS	wt	Extensive positive	Negative	wt	No methylated	Lynch syndrome
20	50	Male	CD	Concomitant	Ileum	NOS with mucinous component	wt	Extensive positive	Negative	wt	NP	Unknown
24	54	Male	CD	1	Ileum	NOS	wt	Focal positive	Negative	wt*	NP	Unknown
53	52	Female	CD	2	Right colon	Mucinous and signet ring cell	wt	Extensive positive	Negative	wt	NP	Unknown

Abbreviations: IBD, Intestinal bowel disease; UC, ulcerative colitis; CD, Crohn’s disease; wt, wild type; IHC, immunohistochemistry; NP, not performed. wt*, in this case a G13D mutation in *KRAS* was identified.

**Table 7 ijms-24-08655-t007:** Expression of MUC5AC in mucosa adjacent to adenocarcinomas.

	MUC5AC			
	Negative (%)	Positive (%)	Total (%)	*p* Value
Adjacent mucosa (*n* = 35)				1.000
Location				
Colon	8 (100)	26 (96)	34 (97)	
Small bowel	0	1 (4)	1 (3)	
Specific location				0.614
Ileum	0	1 (4)	1 (3)	
Right	3 (37.5)	7 (26)	10 (29)	
Left	3 (37.5)	12 (44)	15 (43)	
Rectum	0	4 (15)	4 (11)	
Colon	2 (25)	3 (11)	5 (14)	
Status				0.374
Active inflammation	0	4 (14.8)	4 (11.4)	
Chronic changes	4 (50)	15 (55.6)	19 (54.3)	
Normal	4 (50)	8 (29.6)	12 (34.3)	
MUC5AC in adenocarcinoma				0.700
Negative	4 (50)	16 (59)	20 (57)	
Positive	4 (50)	11 (41)	15 (43)	

## Data Availability

The data that support the findings of this study are available from the corresponding author upon reasonable request.

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
