# Peer review of "Alterations in p53, Microsatellite Stability and Lack of MUC5AC Expression as Molecular Features of Colorectal Carcinoma Associated with Inflammatory Bowel Disease"

_ijms, 2023, doi:10.3390/ijms24108655_

Round 1

Reviewer 1 Report

Gene et al. have comprehensively investigated the expression of p53, markers of the MMR pathway, and MUC5AC for gastric metaplasia. Through immunohistochemical assessments and clinicipathological characterization, the authors have clearly showed that MUC5AC staining is predominantly observed in p53 negative MSI high colitis-associated colorectal carcinoma. I have no corrections in the manuscript and agree to be published in the present form.

Author Response

Response to reviewer 1 comments

Thank you very much.

Reviewer 2 Report

This study characterized colitis-associated colorectal carcinoma, analyzing p53 alterations, microsatellite instability (MSI), and MUC5AC expression. The p53 expression was examined by IHC. So called p53 “mut-pattern” was found in more than half of the CAC, most frequently stable (MSS) and MUC5AC negative. MUC5AC staining was more frequently observed in intestinal mucosa, inflamed or with chronic changes, than in CAC, especially in those with “p53 wt-pattern” and MSS. Based on their results, the authors proposed that in IBD GM occurs in inflamed mucosa, persists in those with chronic changes and disappears with the acquisition of p53 mutations. Several issues need to be addressed to improve the manuscript.

Major

1.     Although abnormal p53 IHC staining patterns may correlated well with its mutation (or null) status, they should not be considered as the same as mutation analyses. As indicated in the cited reference (ref. 39), some cancer tissues with p53 mutations may show “Wild type” staining pattern. The manuscript should be revised to indicate this fact.

2.     The method to examine microsatellite instability (Table 2) was not provided in the Materials and Methods Section. How did the authors define MSI and MSS?

3.     The authors concluded that “In relation to the expression of MUC5AC in CAC and surrounding intestinal mucosa we hypothesized that the appearance of GM occurs in inflamed mucosa, persists in those with chronic-regenerative changes and disappears with the acquisition of p53 mutations.” With limited case numbers of association data obtained by inconclusive methods, this may be an overreaching statement.

Minor

4.     The meaning of the sentence in lines 96-98 was unclear, please revised it.

5.     “NOS adenocarcinomas” in line 101 should be “adenocarcinoma”.

Author Response

Response to Reviewer 2 Comments

Major

  1. Although abnormal p53 IHC staining patterns may correlated well with its mutation (or null) status, they should not be considered as the same as mutation analyses. As indicated in the cited reference (ref. 39), some cancer tissues with p53 mutations may show “Wild type” staining pattern. The manuscript should be revised to indicate this fact.

ANSWER

We appreciate the reviewer for the comment.

In section 4. Material and Methods, the sentence in lines 259-260 has been modified as follows:

The wild-type pattern, corresponding to focal nuclear positivity of varying intensity, was considered normal (p53-wt pattern), even knowing that it could be a false negative for a mutated TP53 [39].

  1. The method to examine microsatellite instability (Table 2) was not provided in the Materials and Methods Section. How did the authors define MSI and MSS?

ANSWER

We appreciate the reviewer for the comment.

In section 4. Materials and Methods, 4.1 Immunohistochemistry, page 12, on lines 246-250, describes the antibodies used to evaluate MSI as follows:

“The primary antibodies used were anti-p53 (clone DO-7, Ventana Medical Systems, Inc., 1910 E. Innovation Park Drive, Tucson, Arizona 85755 USA), anti-MUC5AC (clone MRQ-19, Ventana Medical Systems), anti-MLH1 (clone ES05, Leica Biosystems, Newcastle Upon Tyne, UK), anti-MSH2 (clone 79H11, Leica Biosystems), anti-MSH6 (clone EP49, Leica Biosystems), and anti-PMS2 (clone EP51, Leica Biosystems).”

Lines 250-253 describe the machine used as follows:

“Immunostaining was performed automatically using a Ventana BenchMark ULTRA machine (Roche, Basel, Switzerland) for p53 and MUC5AC and a Bond-III machine (Leica Biosystems) for the MMR antibodies.

On lines 268-273, the location of positive staining for MMR antibodies, the definition of MSI and MSS, the presence of internal positive controls, and the definition of loss of MMR staining are detailed as follows:

“Positive staining for the MMR antibodies were in the nucleus of the neoplastic cells. We considered the tumor stable if nuclear staining was retained in tumor cells for all four MMR antibodies and unstable if one or more of the 4 markers lacked nuclear expression in tumor cells. Nuclear immunoreaction in lymphocytes, normal colonic mucosa, or stromal cells served as the internal anti-MMR positive control. Loss of MMR staining was considered when the nuclei of all neoplastic cells were negative.”

  1. The authors concluded that “In relation to the expression of MUC5AC in CAC and surrounding intestinal mucosa we hypothesized that the appearance of GM occurs in inflamed mucosa, persists in those with chronic-regenerative changes and disappears with the acquisition of p53 mutations.” With limited case numbers of association data obtained by inconclusive methods, this may be an overreaching statement.

ANSWER

We appreciate the reviewer for the comment.

We agree with the reviewer that the series analyzed is limited in size and restricted to neoplasms. Therefore, in line 236-238, at the end of section 3. Discussion, we introduce the following sentence:

“Further studies are needed to explore the expression of MUC5AC in dysplastic epithelia IBD-associated, an intermediate step between colon mucosa with chronic changes and the onset of neoplastic events of CAC.”

And in the section 5. Conclusions, we use the verb hypothesize because we simply formulate a hypothesis that is provisionally established based on our research, the validity of which may or may not be confirmed.

According to the reviewer’s appreciation the last sentence of the section 5. Conclusions has been changed as follows:

“In relation to the expression of MUC5AC in CAC and surrounding intestinal mucosa, we could hypothesize that the appearance of GM occurs in inflamed mucosa, persists in those with chronic-regenerative changes and disappears with the acquisition of p53 mutations.”

Minor

  1. The meaning of the sentence in lines 96-98 was unclear, please revised it.

ANSWER

We appreciate the reviewer for the comment.

The sentence has been changed according the reviewer’s appreciation as follows:

“According to location, CAC in patients with CD were more frequently located in the ileum (7/19, 37%) and the right colon (6/19, 31.5%), whereas CAC in patients with UC were more frequent in the left colon (15/38, 39.5%)”.

  1. “NOS adenocarcinomas” in line 101 should be “adenocarcinoma”.

ANSWER

We appreciate the reviewer for the comment.

The sentence has been changed according the reviewer’s appreciation as follows:

“Regarding histology, the majority of CRC (47/57, 82.5%) were adenocarcinomas in both CD and UC”.

Reviewer 3 Report

In this report, the authors present expression profiles of various genes including p53 in CAC and other cancers with the goal to understand the molecular progression of these cancers. The data have some value but somewhat minimal considering that the genetic signatures of these cancers have been identified. Why do the authors elect to perform staining rather than mutation or gene expression analysis which is more common now? The value of these methods is not well described in the introduction. Clearly, the authors find some interesting results that they elegantly list in the conclusion section so the introduction should dedicate a paragraph to why staining is preferred over other methods. Is it faster, cheaper, etc…

Furthermore, the authors sometimes use the word “mutation” as in p53-mut but they do not characterize mutation. Care should be taken to indicate whether gene expression or mutation (or both) is analyzed. AS far as I can tell, no actual mutation analysis is performed. Unless the samples were already known to harbor various mutations. If so, this should be stated. Take, for example, table 4. When the authors say “mut-pattern” vs “wt-pattern” are they referring to p53 sequence alterations or gene expression changes? This is important because changes in gene expression does not necessarily equal sequence alterations (e.g. mutation).

Other careless errors are made in figure labeling, various size fonts, etc. Please improve the presentation.

Major revisions

1.      The authors refer to “p53 mutated pattern” but they only check this by staining which looks at gene expression levels and not mutation (line 108). Are the authors checking DNA sequence or only looking at gene expressions? The authors should sequence the p53 gene if they intend to check mutation. Otherwise, please refine your writing to indicate that only gene expression patterns were investigated. The way it is written now is confusing.

2.      At line 154 the authors describe a BRAF V600E activating mutation in some patients. How do the authors know this? Did they sequence this patient? And if so, why did they not sequence the other genes, e.g. p53, MUC5AC, etc?

3.      The paper is written somewhat disorganized and it is hard to follow some of the experiments that the authors list. I suggest that a diagram be made to accompany the conclusion paragraph. Such a diagram would be highly informative particularly for readers not familiar with these cancers or the experiments described here.

Minor revisions

1.      Line 51 abstract. It may be better to say “based on our results we conclude…” rather than “hypothesize” since technically your results lead to a conclusion.

2.      What do the authors mean by this?  “The sequence “chronic inflammation-dysplasia-cancer” should determine the driving genes that guide this carcinogenesis pathway of CRC, different from the canonical and serrated pathways.” Perhaps you mean, “the sequence of events…”. The way it is written is somewhat awkward.

3.      Table 1 should be labeled better. For example, the row “age” has no numbers in it but then “mean years” has a range. Are the Age/Mean years meant to be the same row? The way it is formatted now is a little hard to interpret.  Perhaps include another column such as in table 2 if labeling of some rows take too much space. Also, what is meant by concominant? Do you mean “concomitant”? And if you mean “concomitant” what does that refer to? I suggest that footnotes should be created under the table to explain these labels. Additionally, please pay attention to the numbers. For example, p-values are sometimes listed as 0.780 and other times 0,042. Please choose either comma or period.  

4.      Figure labels are also non-standard. Sometimes a), b) etc, is used and other times A, B, C, etc. Please stick with one or the other. Also pay attention to the fonts and the figure legends to make sure that if A is used to label a panel, A is also used in the legend.

Author Response

Response to Reviewer 3 Comments

Major revisions

  1. The authors refer to “p53 mutated pattern” but they only check this by staining which looks at gene expression levels and not mutation (line 108). Are the authors checking DNA sequence or only looking at gene expressions? The authors should sequence the p53 gene if they intend to check mutation. Otherwise, please refine your writing to indicate that only gene expression patterns were investigated. The way it is written now is confusing.

ANSWER

We appreciate the reviewer for the comment.

In section 4. Materials and Methods, 4.1 Immunohistochemistry, page 12, on lines 254-260, describes that we evaluated p53 immunostaining using the 3-level scores described by Köbel et al. as follows:

“For the evaluation of p53 immunostaining we used the 3-level scores described by Köbel et al. that correlated with TP53 alterations: the overexpression pattern, when strong and intense p53 staining was observed in all nuclei, and the complete absence (null pattern) when a total absence of nuclear staining was observed were considered as surrogates for TP53 mutation (p53-mut pattern) [39]. The wild-type pattern, corresponding to focal nuclear positivity of varying intensity, was considered normal (p53-wt pattern) (Figure 5). An external positive control was included on each slide.”

In section 2. Results, 2.2. Immunohistochemistry, page 4, lines 108-111 and Table 2, we refer to the different p53 expression patterns defined in section 4. Material and Methods.

As the reviewer comments, no molecular study has been performed to identify TP53 gene mutations. The order in which the different sections of the article are arranged is given by the publisher.

To facilitate the understanding of the results of p53 expression we introduce the following changes in line 108:

The p53-mut pattern (defined in Material and Methods) was found in more than half (29/57, 51%) of the adenocarcinomas (22 with overexpression and 7 with null pattern).

  1. At line 154 the authors describe a BRAF V600E activating mutation in some patients. How do the authors know this? Did they sequence this patient? And if so, why did they not sequence the other genes, e.g. p53, MUC5AC, etc?

ANSWER

We appreciate the reviewer for the comment.

Analysis of BRAF mutational status, as a surrogate for hypermethylated MLH1, is recommended in MSI-H MLH1-negative colorectal carcinomas to guide the study of Lynch syndrome. Therefore, the BRAF V600E mutation and MLH1 methylation data were obtained from the review of medical records.

Sequencing the p53 and MUC5AC genes is beyond the scope of the budget of this study.

  1. The paper is written somewhat disorganized and it is hard to follow some of the experiments that the authors list. I suggest that a diagram be made to accompany the conclusion paragraph. Such a diagram would be highly informative particularly for readers not familiar with these cancers or the experiments described here.

ANSWER

We appreciate the reviewer for the comment.

As we comment, the order in which the different sections of the article are arranged is given by the publisher.

Minor revisions

  1. Line 51 abstract. It may be better to say “based on our results we conclude…” rather than “hypothesize” since technically your results lead to a conclusion.

ANSWER

We appreciate the reviewer for the comment.

Line 51 abstract has been modified following the reviewer’s recommendation as follows:

“Based on our results we conclude that, as in the serrated pathway of CRC, in IBD GM occurs in inflamed mucosa, persists in those with chronic changes and disappears with the acquisition of p53 mutations.”

  1. What do the authors mean by this? “The sequence “chronic inflammation-dysplasia-cancer” should determine the driving genes that guide this carcinogenesis pathway of CRC, different from the canonical and serrated pathways.” Perhaps you mean, “the sequence of events…”. The way it is written is somewhat awkward.

ANSWER

We appreciate the reviewer for the comment.

Colorectal carcinoma arising in inflammatory bowel disease is considered the prototype of inflammation-induced carcinogenesis. We use the terms "chronic inflammation-dysplasia-cancer" because it simplifies the events that occur in the pathogenesis of this type of colorectal carcinoma, as other authors have done previously (Wang CP, Lin JJ, Shah SC, Kim MK; Colorectal Cancer Screening Working Group. Geographic Variation in Colorectal Cancer Incidence Among Asian Americans: A Population-Based Analysis 2006-2016. Clin Gastroenterol Hepatol. 2023 Feb;21(2):543-545.e3. doi: 10.1016/j.cgh.2022.01.026. Epub 2022 Feb 3. PMID: 35123087; PMCID: PMC9346101., page 716, line 13: “Markers of oxidative damage and DNA double strand breaks increase progressively in the inflammation-dysplasia-carcinoma sequence.”).

  1. Table 1 should be labeled better. For example, the row “age” has no numbers in it but then “mean years” has a range. Are the Age/Mean years meant to be the same row? The way it is formatted now is a little hard to interpret. Perhaps include another column such as in table 2 if labeling of some rows take too much space. Also, what is meant by concominant? Do you mean “concomitant”? And if you mean “concomitant” what does that refer to? I suggest that footnotes should be created under the table to explain these labels. Additionally, please pay attention to the numbers. For example, p-values are sometimes listed as 0.780 and other times 0,042. Please choose either comma or period. 

ANSWER

We appreciate the reviewer for the comment.

In IBD duration “Concomitant” means that the IBD diagnosis was made at colectomy. An explanation has been added in the table 1 footnotes.

According to reviewer’s comments we have modified Table 1 as follows:

Table 1. Clinicopathological features of IBD patients and adenocarcinomas.

Crohn’s

disease (%)

Ulcerative

colitis (%)

Total

(%)

P value

Patients

19 (36)

34 (64)

53 (100)

Age Mean years [range]

58.4 [35-88]

64.8 [34-90]

62.6 [34-90]

Gender

1.000

Male

14 (74)

24 (71)

38 (72)

Female

5 (26)

10 (29)

15 (28)

IBD duration

0.244

Mean years [range]

13.2 [1-43]

Concominant*

5 (26)

2 (6)

7 (13.2)

≤8 years

4 (21)

13 (38)

17 (32.1)

9-15 years

3 (16)

7 (20.6)

10 (18.9)

16-20 years

1 (5)

5 (14.7)

6 (11.3)

>21 years

5 (26)

6 (17.6)

11 (20.8)

Unknown

1 (5)

1 (3)

2 (3.8)

IBD activity

0.780

Active

11 (58)

18 (53)

29 (55)

Quiescent

8 (42)

16 (47)

24 (45)

Synchronous CRC (n = 57)

0.042

Non synchronous CRC

19 (100)

30 (79)

49 (86)

2 synchronous CRC

0

8 (21)

8 (14)

Adenocarcinoma location (n = 57)

0.003

Ileum

7 (37)

0

7 (12.3)

Right colon

6 (31.5)

11 (28.9)

17 (29.8)

Left colon

3 (16)

15 (39.5)

18 (31.6)

Rectum

1 (5)

6 (15.8)

7 (12.3)

Colon

2 (10.5)

5 (13.2)

7 (12.3)

Ileal reservory

0

1 (2.6)

1 (1.8)

Histology (n = 57)

0.504

Adenocarcinoma NOS

15 (78.9)

32 (84.2)

47 (82.5)

Mucinous

3 (15.8)

4 (10.5)

7 (12.3)

Mucinous and signet ring cell

1 (5.3)

0

1 (1.8)

Medullary

0

1 (2.6)

2 (1.8)

Undifferentiated

0

1 (2.6)

1 (1.8)

Histologic grade (n = 57)

1.000

Low-grade

15 (79)

31 (82)

46 (81)

High-grade

4 (21)

7 (18)

11 (19)

Adjacent mucosa to

adenocarcinoma (n = 57)

0.077

Absent

10 (53)

12 (32)

22 (39)

Present

9 (47)

26 (68)

35 (61)

Active inflammation

0

4 (15)

4 (11.4)

Chronic changes

4 (21)

15 (58)

19 (54.3)

Normal

5 (26)

7 (27)

12 (34.3)

Stage (8th ed) (n = 57)

0.305

I

3 (15.8)

9 (23.7)

12 (21)

II

6 (31.6)

17 (44.7)

23 (40)

III

10 (52.6)

12 (31.6)

22 (39)

Abbreviations: IBD, intestinal bowel disease; NOS, no otherwise specified. Concomitant*, in this case the diagnosis of IBD was made at colectomy.

  1. Figure labels are also non-standard. Sometimes a), b) etc, is used and other times A, B, C, etc. Please stick with one or the other. Also pay attention to the fonts and the figure legends to make sure that if A is used to label a panel, A is also used in the legend.

ANSWER

We appreciate the reviewer for the comment.

Following the reviewer's recommendation, we have removed A, B, C from Figure 3 and changed them to labels. In addition, we have removed A, B, C from the figure legend as follows:

Figure 3. Colon mucosa adjacent to a CAC. A) Severe distortion of colonic epithelia with chronic inflammation, glandular atrophy, and granulation tissue (H&E, 10x). B) Strong expression of MUC5AC was seen in the epithelium re-epithelizing the granulation tissue and in part of the irregular glands. C) p53-wt pattern showed few positive nuclei in the surface epithelium and in the glands (in magnification, 40x).

Reviewer 4 Report

In this manuscript, aim of the study is  to characterize colitis-associated colorectal carcinoma analyzing p53 alterations and microsatellite instability and to explore their relation to gastric metaplasia using a total of only 57 patients. In the results author showed clinicopathological features of patients, IHC-Expression of p53, MSI and MUC5AC in patients and relation between the markers where these results conclude that CAC shows a molecular profile characterized by p53 alterations and absence of both MSI and MUC5AC staining that differentiate  from CRC of the canonical and serrated carcinogenesis pathways.  Manuscript is well written with relevant  details in all the sub headings. However this manuscript have a too small sample size, missing patients ethnicity information and IHC images need scale bar. I would not recommend to IJMS because of  small sample size. 

Author Response

Response to Reviewer 4 Comments

We appreciate the reviewer for their comments.

  1. Small sample size

We agree with the reviewer that the series analyzed is limited in size and, also, restricted to neoplasms. Therefore, in line 236-238, at the end of section 3. Discussion, we introduce the following sentence:

“Further studies are needed to explore the expression of MUC5AC in dysplastic epithelia IBD-associated, an intermediate step between colon mucosa with chronic changes and the onset of neoplastic events of CAC.”

And in the section 5. Conclusions, we use the verb hypothesize because we simply formulate a hypothesis that is provisionally established based on our research, the validity of which may or may not be confirmed.

According to the reviewer’s appreciation the last sentence of the section 5. Conclusions has been changed as follows:

“In relation to the expression of MUC5AC in CAC and surrounding intestinal mucosa, we could hypothesize that the appearance of GM occurs in inflamed mucosa, persists in those with chronic-regenerative changes and disappears with the acquisition of p53 mutations.”

  1. Missing patients ethnicity information

In accordance with the missing patient ethnicity information, we have reviewed the medical records and added this information in section 2. Results, page 2, lines 94-95 as follows:

“A total of 57 CAC were studied from 38 (72%) men and 15 (28%) women, all Caucasian, aged between 34 and 90 years (mean 62.6), 34 (64%) with UC and 19 (36%) with CD.”

  1. IHC images need scale bar

We have added the scale bar to IHC images 1, 2, 3 and 5 and changed all figure legends as follows (in the manuscript with changes it was not possible to introduce the new images shown):

Figure 1. Two CACs with different expression of p53 and MUC5AC. a) p53 shows a wt-pattern with focal nuclear staining in tumor cells. In the same tumor, MUC5AC expression is diffuse (20x, scale bar 50 µm). b) In this other tumor, nuclear overexpression is observed in the form of mut-pattern staining of p53 while MUC5AC is negative. Note the MUC5AC positivity in non-dysplastic glands on the left side (10x, scale bar 100 µm).

Figure 2. Two examples of MSI-H CAC. a) medullary and b) mucinous. In both adenocarcinomas p53 showed scattered nuclear positivity corresponding to a wt-pattern, MUC5AC expression was focal positive, and MLH1 showed absence of nuclear staining in tumor cells. Note in both MLH1 staining the positive internal control in lymphocytes (20x, scale bar 50 µm).

Figure 3. Colon mucosa adjacent to a CAC. A) Severe distortion of colonic epithelia with chronic inflammation, glandular atrophy, and granulation tissue (H&E, 10x, scale bar 100 µm). B) Strong expression of MUC5AC was seen in the epithelium re-epithelizing the granulation tissue and in part of the irregular glands (10x, scale bar 100 µm). C) p53-wt pattern showed few positive nuclei in the surface epithelium and in the glands (10x, scale bar 100 µm; in magnification, 40x).

Figure 4. Colonic mucosa with chronic changes adjacent to a CAC. a) p53 staining showed a wt-pattern with few positive nuclei (in magnification) in the colon glands adjacent to an adenocarcinoma with p53 nuclei overexpression (p53 mut-pattern) (10x, scale bar 100 µm). b) MUC5AC strongly positive in the colon mucosa adjacent to a negative neoplasm (5x, scale bar 200 µm).

Figure 5. Three patterns of p53 staining. a) Overexpression pattern, all tumor nuclei are strongly positive (scale bar 10 µm). b) Null pattern, no staining is observed in tumor cell nuclei in the presence of a positive internal control in stromal cells (scale bar 10 µm). c) Wild-type pattern, focal nuclear positivity of varying intensity in tumor cell nuclei (scale bar 50 µm).

Round 2

Reviewer 2 Report

The revision has addressed comments in the previous review.

Author Response

Thank you very much.

Reviewer 3 Report

The authors have made significant changes or provided compelling arguments of them. The paper is much improved. However, I still think that a concluding diagram should be introduced at the end to summarize the findings. Perhaps "disorganized" was not the right word. What I meant was that there are a lot of data presented and is a little hard to follow. A concluding diagram would be great.

Author Response

Response to Reviewer 3 Comments

Minor changes

The authors have made significant changes or provided compelling arguments of them. The paper is much improved. However, I still think that a concluding diagram should be introduced at the end to summarize the findings. Perhaps "disorganized" was not the right word. What I meant was that there are a lot of data presented and is a little hard to follow. A concluding diagram would be great.

ANSWER

We appreciate the reviewer for the comment.

Following the reviewer’s recommendation, we have added at the end of section 5 Conclusion Figure 6 and the figure legend as follows:

Figure 6. Pathways of carcinogenesis in IBD. GM appears in the inflamed colic mucosa as an adaptive mechanism to continued stress, persists in the mucosa with chronic changes and disappears with the acquisition of p53 mutations. CAC are characterized by p53 alterations, lack of MUC5AC expression and MSS, whereas those CRC with MSI-H, MUC5AC expression and p53 wt-pattern could represent sporadic CRC of the serrated pathway that appear in a context of "chronic inflammation-dysplasia-cancer" from a gastric metaplastic epithelium. Figure created with BioRender.com.

Reviewer 4 Report

author response has been accepted. subject/content of the manuscript is not IJMS standard. Editor may take decision to publish in this journal or other journal in MDPI 

Author Response

Thank you very much.